# AesFormer: Transform Everyday Photos into Beautiful Memories

**Tianxiang Du** [1]  **Hulingxiao He** [1]  **Yuxin Peng** [1]

## Abstract

In everyday photography, aesthetically appealing moments are often captured with structural flaws (e.g., composition, camera viewpoint, or pose) that existing retouching and portrait enhancement methods cannot fix. We formulate **Aesthetic Photo Reconstruction (APR)** as improving a photo's aesthetic quality via structural reconstruction while preserving subject identity and scene semantics. Although recent advances in image editing models make APR feasible, they often lack aesthetic understanding, yielding edits that are semantically plausible yet aesthetically weak. To address this, we propose **AesFormer**, a two-stage framework that decouples aesthetic planning from image editing. In Stage 1, an aesthetic action model (**AesThinker**) analyzes the input along seven progressive photographic dimensions and outputs executable editing actions; we further apply **GRPO-A** to encourage broad exploration over diverse action plans beyond SFT. In Stage 2, an action-conditioned editor (**AesEditor**) performs structural edits guided by these actions. To support APR, we build a video-based corpus-mining pipeline (VCMP) and construct **AesRecon**, a benchmark of 9,071 strictly aligned (poor, good) image pairs. Experiments show that AesFormer substantially improves APR performance and is competitive with Nano Banana Pro. Code is available at https://github.com/PKU-ICST-MIPL/AesFormer_ICML2026.

## 1. Introduction

Photography is a common way to preserve everyday scenes, emotions, and memories (Duan et al., 2025; Lin et al., 2025). Yet truly compelling moments are fleeting: capturing them

[1]Wangxuan Institute of Computer Technology, Peking University, Beijing, China. Correspondence to: Yuxin Peng <pengyuxin@pku.edu.cn>.

*Proceedings of the 43rd International Conference on Machine Learning*, Seoul, South Korea. PMLR 306, 2026. Copyright 2026 by the author(s).

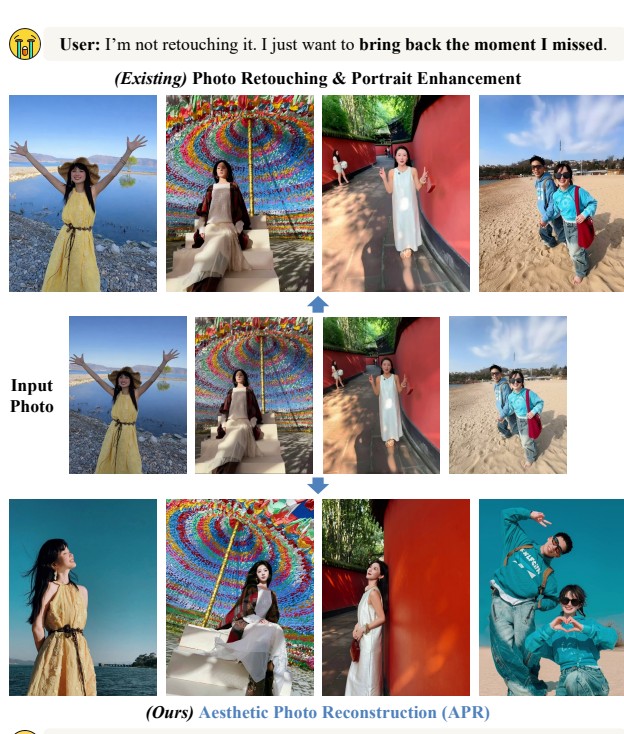

**User:** I'm not retouching it. I just want to **bring back the moment I missed**.

*(Existing)* Photo Retouching & Portrait Enhancement

Input Photo

*(Ours)* Aesthetic Photo Reconstruction (APR)

**User:** That's beautiful. It feels **exactly like what I was trying to capture**.

*Figure 1.* Conventional photo retouching and portrait enhancement mainly improve style and appearance but cannot fix structural flaws introduced at capture, whereas our **APR** performs aesthetic-driven structural reconstruction to bring back the intended moment.

well often requires split-second decisions on framing, camera viewpoint, and subject pose at the moment of shooting (Jiang et al., 2022; Liu et al., 2025b; Li et al., 2025b). Professional photographers develop this ability through systematic training and extensive practice, whereas most users struggle to make these decisions reliably (Lou et al., 2021; Li et al., 2025b). As a result, their photos frequently contain structural flaws—unbalanced framing (e.g., a misplaced subject or distracting background), an ill-chosen viewpoint that weakens depth or introduces distortion, or stiff poses that dilute the intended emotion (Lou et al., 2021; Zhao et al., 2025; Du et al., 2026). These capture-time errors create a persistent gap between the moment users intend to preserve and what the camera ultimately records.

To narrow this gap, users often resort to post-processing tools to improve photo aesthetics. Existing solutions largely

fall into two categories. Photo retouching methods (Bychkovsky et al., 2011; Liang et al., 2021; Ouyang et al., 2023; Dutt et al., 2025; Lin et al., 2025) and tools such as Photoshop and Lightroom typically adjust low-level attributes (e.g., brightness, contrast, and saturation) to refine overall tone and style. Portrait enhancement methods (Xie et al., 2023; Jin et al., 2024; Xu et al., 2025) focus on appearance-oriented edits, including skin smoothing and facial refinement. However, these tools are not designed to correct structural mistakes made at capture time. As illustrated in Figure 1, when composition, viewpoint, or pose is suboptimal, style/appearance edits provide only limited gains and rarely achieve the intended aesthetic outcome.

This limitation highlights an under-explored problem: improving a photo's aesthetic quality via structural reconstruction by adjusting composition, camera viewpoint, subject placement and pose, and other structural attributes, while preserving subject identity and scene semantics. We formalize this task as **Aesthetic Photo Reconstruction (APR)**, aiming to bridge the aesthetic gap between the intended moment and the photo actually captured by the camera.

Recent advances in image editing models (Li et al., 2025a; Deng et al., 2025; Labs et al., 2025; Liu et al., 2025a; Wu et al., 2025a) make APR increasingly feasible by enabling instruction-guided edits. However, current models still face two key limitations. **(1) Limited aesthetic understanding.** They often lack a robust grasp of photographic aesthetics, making it difficult to diagnose structural issues and decide how to correct them. **(2) Limited aesthetic editing.** Even with explicit instructions, the resulting edits are often semantically plausible yet aesthetically weak, failing to deliver clear and reliable aesthetic gains.

To address these challenges, we propose **AesFormer**, a two-stage framework that decouples aesthetic understanding from image editing. In Stage 1, an aesthetic action model, **AesThinker**, analyzes the input photo across seven progressive photographic dimensions and outputs executable editing actions. We first apply supervised fine-tuning (SFT) to standardize the action format, then introduce **GRPO-A** (Group Relative Policy Optimization for Aesthetics) to encourage diverse, high-quality action plans, reflecting the multi-solution nature of aesthetics. In Stage 2, an action-conditioned editor, **AesEditor**, executes these actions to perform photo reconstruction via structural edits.

Another major obstacle for APR is data: high-quality paired examples of the same scene and subject are scarce, making it hard to learn reliable structural corrections (You et al., 2025). To address this, we propose a video-based corpus-mining pipeline (**VCMP**) that extracts before/after demonstrations from photography tutorial videos and applies strict refinement and alignment. With VCMP, we semi-automatically build **AesRecon**, a new dataset and benchmark containing 9,071 strictly aligned (poor image, good image) pairs.

Experiments show that AesFormer substantially improves APR, outperforming open-source baselines and achieving competitive performance against Nano Banana Pro.

The main contributions are summarized as follows:

- **Aesthetic Photo Reconstruction (APR)** is introduced as a new task that improves photo aesthetics via structural reconstruction, beyond conventional retouching.

- We propose **AesFormer**, a two-stage framework that decouples aesthetic understanding from image editing, and further introduce **GRPO-A** to encourage broad exploration over diverse action plans, achieving results competitive with Google's Nano Banana Pro.

- To address the data bottleneck in APR, we develop a video-based corpus-mining pipeline (**VCMP**) and construct **AesRecon**, an APR dataset and benchmark of 9,071 strictly aligned(poor, good) image pairs mined from photography tutorial videos.

## 2. Related Work

**Retouching and Portrait Enhancement.** Prior work on improving photo aesthetics through post-processing largely falls into two lines. Photo retouching studies learn global light and color adjustments, refining low-level attributes such as exposure, contrast, and white balance (Bychkovsky et al., 2011; He et al., 2020; Liang et al., 2021; Ouyang et al., 2023; Duan et al., 2025; Dutt et al., 2025; Lin et al., 2025). In parallel, portrait enhancement focuses on appearance-centric refinements (e.g., skin smoothing, facial reshaping, and detail enhancement), often with specific architectures and datasets (Xie et al., 2023; Jin et al., 2024; Xu et al., 2025). Despite strong perceptual gains, these methods mainly operate in the tonal/appearance space. When a photo suffers from capture-time structural issues (e.g., a misplaced or partially clipped subject, a tilted horizon, or a stiff pose), such adjustments are often insufficient to recover the desired aesthetics, which motivates APR.

**Image Editing.** The rise of diffusion models has reshaped text-to-image (T2I) synthesis (Rombach et al., 2022; Lipman et al., 2022; Yan et al., 2025) and greatly advanced image editing. Existing methods typically condition on explicit textual instructions to perform global or local content/style modifications. Recent work further moves toward instruction-tuned editors (Liu et al., 2025a; Labs et al., 2025; Zhang et al., 2025; Li et al., 2025a) and unified multimodal models (Li et al., 2025d; Wu et al., 2025b; Deng et al., 2025), alongside progress in foundational architectures such as flow matching (Lipman et al., 2022). These methods primarily target instruction following and semantic

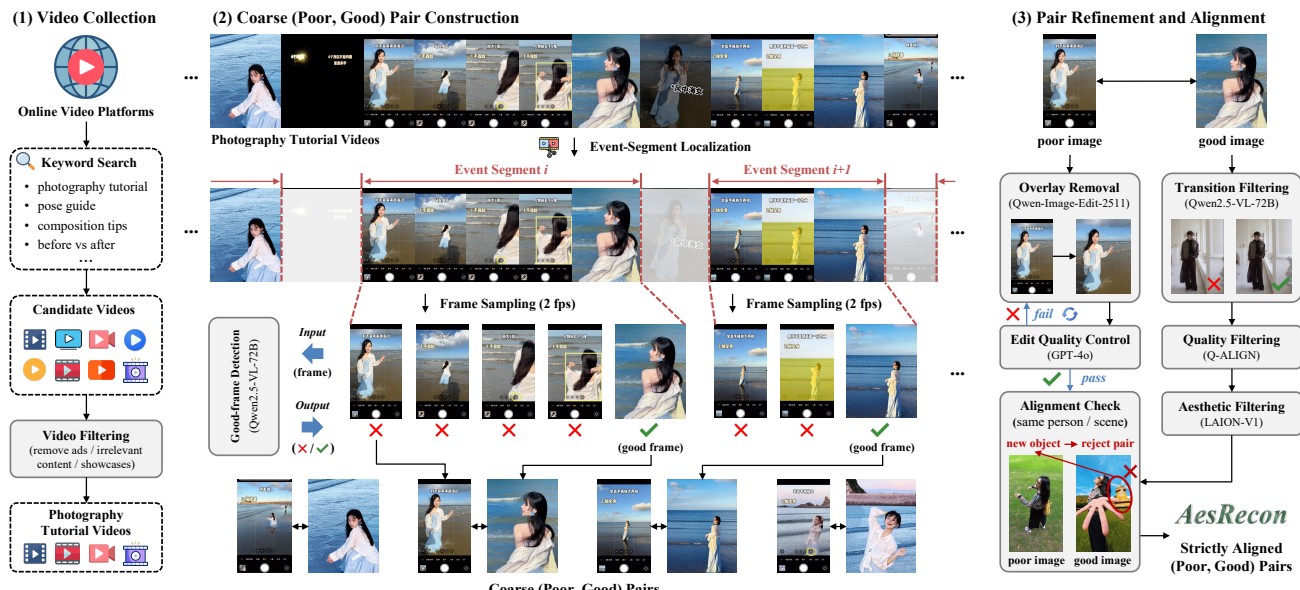

*Figure 2.* Overview of the proposed video-based corpus-mining pipeline (**VCMP**). It collects photography tutorial videos from online video platforms, constructs coarse (poor, good) pairs from before/after demonstrations, and refines them via multi-stage filtering and strict alignment, resulting in **AesRecon**, a new APR dataset and benchmark with 9,071 strictly aligned (poor, good) image pairs.

consistency, and are often evaluated by instruction–output alignment (Ye et al., 2025; Liu et al., 2025a). However, this paradigm falls short for APR: rather than fulfilling an explicit request, APR aims to improve overall aesthetics by correcting capture-time structural errors (e.g., in composition, viewpoint, and pose). In this setting, the model must diagnose aesthetic issues and decide how to improve them via structural changes, which remains challenging for off-the-shelf instruction-based editors (You et al., 2025).

## 3. VCMP and the AesRecon Benchmark

Aesthetic Photo Reconstruction (APR) is highly data-constrained. It requires strictly aligned (poor, good) image pairs that depict the same subject and scene, where the aesthetic improvement is primarily attributable to structural factors (e.g., improved framing and more natural pose), rather than tone/color adjustments. Such aligned pairs are exceedingly scarce in existing image collections.

Online photography tutorial videos offer a scalable source of supervision. A typical tutorial records multiple shooting events in which the photographer and subject iteratively refine a shot, progressing from an initial suboptimal capture to a final high-quality photo. Along the video timeline, each event usually forms a temporally coherent segment containing both before and after demonstrations.

However, reliably mining pairs from raw videos is non-trivial due to ads or sponsored content, transition artifacts (e.g., blur and ghosting), and pervasive screen overlays (e.g., camera UI and subtitles). To address these challenges, we

propose a video-based corpus-mining pipeline (**VCMP**) that systematically converts raw tutorial videos into strictly aligned (poor, good) pairs, as shown in Figure 2. Using VCMP, we build **AesRecon**, a new APR dataset and benchmark comprising 9,071 strictly aligned pairs.

### 3.1. Video Collection

We retrieve 5,700 candidate videos from online video platforms (e.g., Rednote, TikTok, and YouTube) using photography-instruction keywords (e.g., photography tutorial, pose guide, composition tips, and before vs. after). We first deduplicate videos by their IDs, and then conduct video-level screening to exclude non-instructional content, including ads/sponsored videos, irrelevant videos, and showcase-only clips that lack step-by-step demonstrations. This process yields 2,144 tutorial videos, denoted as $\mathcal{V}$.

### 3.2. Coarse (Poor, Good) Pair Construction

Given a tutorial video $v \in \mathcal{V}$, we extract coarse (poor, good) frame pairs within each shooting event. The procedure consists of two steps: **(1) Event-segment Localization** and **(2) Good-frame Detection**.

**Event-segment Localization.** Tutorial videos typically interleave informative shooting events with low-value footage, such as narration-only shots, interludes, and transitions. We first localize a set of event segments $\mathcal{S}(v) = \{s_k\}_{k=1}^{K}$ from each tutorial video. Each segment $s_k$ is a continuous interval with timestamps $(\tau_k^{\text{start}}, \tau_k^{\text{end}})$, intended to capture a single coherent shooting event where the subject and scene remain

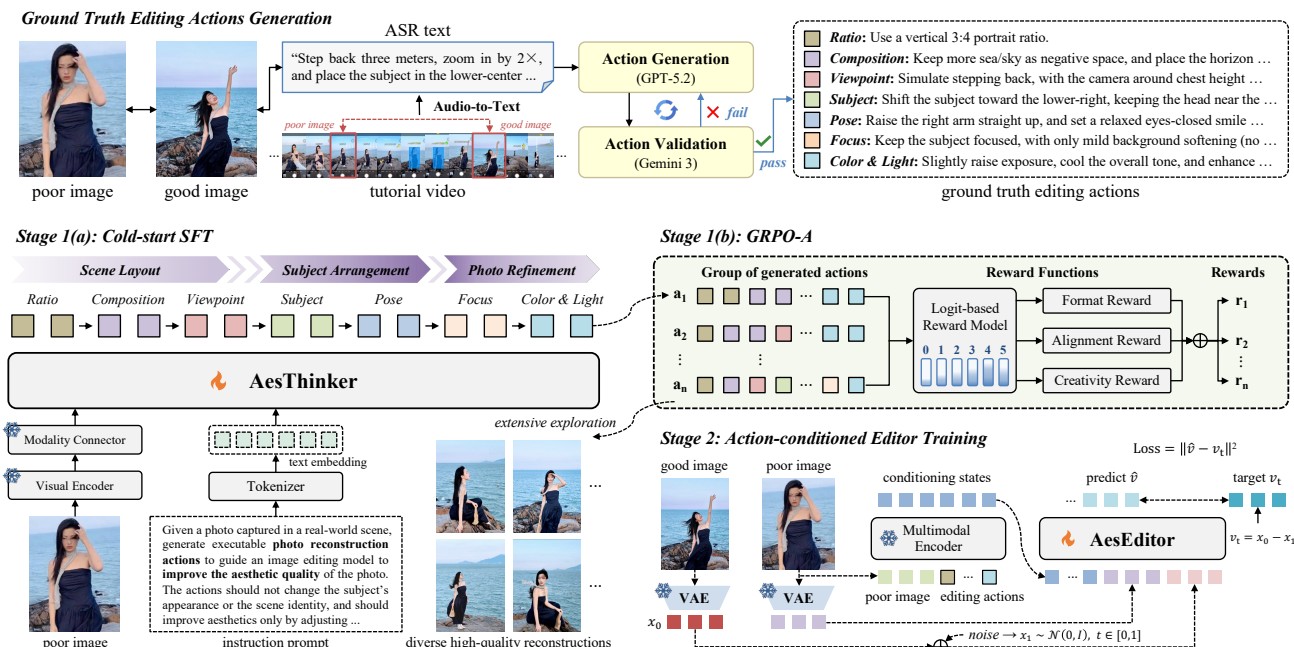

*Figure 3.* Overview of the **AesFormer** framework. Stage 1 trains an aesthetic action model (**AesThinker**) to produce executable, ordered editing actions across seven progressive photographic dimensions. AesThinker is cold-started with SFT using ground-truth actions distilled from tutorial videos, and further optimized with **GRPO-A** to encourage broad exploration over diverse action plans. Stage 2 trains an action-conditioned editor (**AesEditor**) to reconstruct the photo by executing the predicted actions via structural edits.

largely unchanged. We then uniformly sample frames from each segment at 2 fps, i.e., one frame every 0.5 seconds, yielding a frame set $\mathcal{F}(s_k) = \{x_t\}$.

**Good-frame Detection.** Within an event segment, most sampled frames come from the teaching process and thus contain screen overlays (e.g., subtitles, annotations, or camera UI). The final result, however, is usually shown as a clean, photo-like frame. Motivated by this pattern, we use a pretrained vision-language model (VLM), Qwen2.5-VL-72B (Bai et al., 2025), to classify each sampled frame $x_t \in \mathcal{F}(s_k)$ as *good* or *non-good*. For each detected good frame $g = x_{t^*}$ in segment $s_k$, we take the first sampled frame from $s_k$ as the corresponding poor image $p$, forming a coarse pair $(p, g)$. This design follows the typical tutorial narrative: the segment begins with an initial baseline shot and later presents the improved final result, yielding a natural poor→good correspondence.

### 3.3. Pair Refinement and Alignment

**Good-image Filtering.** We first filter out pairs whose good image $g$ is visually low-quality or aesthetically weak. Specifically, we apply an image-quality scorer (Wu et al., 2023) and an aesthetic scorer (Schuhmann et al., 2022) to remove such $g$. We further use Qwen2.5-VL-72B (Bai et al., 2025) to detect transition artifacts (e.g., motion blur and ghosting) and discard affected samples. The remaining candidates form a cleaner set, denoted as $\mathcal{P}_1$.

**Overlay Removal.** Poor images $p$ are typically extracted from instructional frames and often contain screen overlays (e.g., subtitles, icons, camera assist lines, and UI widgets), which are undesirable for APR training. We apply Qwen-Image-Edit-2511 (Wu et al., 2025a) to remove these overlays, producing a cleaned poor image $\tilde{p}$. Since automated overlay removal may fail or inadvertently alter underlying content, we introduce an edit quality-control step: GPT-4o (Hurst et al., 2024) verifies that overlays are removed while the scene and subject identity remain unchanged. If verification fails, we re-edit and iterate until it passes. The resulting refined pairs constitute $\mathcal{P}_2 = \{(\tilde{p}, g)\}$.

**Strict Alignment.** We enforce strict alignment between $\tilde{p}$ and $g$. Concretely, Qwen2.5-VL-72B (Bai et al., 2025) checks whether each pair depicts the same person and the same scene, and originates from the same shooting event. We reject pairs with (i) substantial scene changes, (ii) identity inconsistency, or (iii) newly introduced objects or scene elements in $g$ that are absent in $\tilde{p}$. The remaining set $\mathcal{P}_{\text{final}}$ forms **AesRecon**, comprising 9,071 strictly aligned (poor, good) image pairs. We sample 903 pairs to construct the test set and use the remaining pairs for training.

## 4. Method

In this section, we present **AesFormer** (Figure 3), a two-stage framework that decouples aesthetic understanding

from image editing. Stage 1 learns **AesThinker** to generate ordered, executable editing actions over seven progressive photographic dimensions, initialized with SFT using tutorial-video–distilled supervision and further optimized with **GRPO-A** for diverse action exploration. Stage 2 trains **AesEditor** to perform action-conditioned photo reconstruction by executing these actions as structural edits.

## 4.1. Stage 1(a): Cold-start SFT

To cold-start AesThinker, we distill ground-truth editing actions from AesRecon and use them as supervision for SFT. Each AesRecon sample contains a strictly aligned pair $(p, g)$ mined by our corpus-mining pipeline (Section 3): $p$ is an aesthetically suboptimal *poor* photo and $g$ is its higher-quality *good* counterpart. When available, we also extract the associated audio segment and transcribe it with ASR to obtain a text cue $t$, which provides complementary intent and contextual information. Conditioned on $(p, g, t)$, we prompt GPT-5.2 (Singh et al., 2025) to produce an ordered, executable action plan describing how to edit $p$ toward $g$.

A key design choice is to represent editing actions as an ordered sequence of photographic decisions. In real workflows, photographers typically move from global to local: aspect ratio $\rightarrow$ framing and composition $\rightarrow$ camera viewpoint $\rightarrow$ subject placement $\rightarrow$ subject pose and action details $\rightarrow$ focus and depth-of-field $\rightarrow$ color and light. While these decisions are largely separable, they exhibit a clear unidirectional dependency (e.g., focus relationships are ill-posed before subject placement is fixed). This ordering stabilizes the planning process and yields a decomposable action space. Accordingly, we instruct GPT-5.2 (Singh et al., 2025) to output actions along these seven progressive dimensions.

To improve reliability, we incorporate an action validation loop. For each generated action plan, we ask Gemini 3 to verify (1) completeness and consistency with the paired images $(p, g)$, and (2) format compliance, including the required seven-dimension ordering. Failed cases receive feedback and are sent back to GPT-5.2 (Singh et al., 2025) for regeneration. The validated editing actions provide structured supervision that explicitly specifies how to transform $p$ into $g$. We further introduce seven pairs of dimension-specific special tokens to delimit each dimension, add them to the vocabulary of AesThinker, and perform SFT (Liu et al., 2023) on AesRecon. Each training sample is represented as $(p, q, a)$, where $p$ is the input *poor* image, $q$ is an instruction prompt, and $a$ is the target editing action sequence. The training objective is to maximize the conditional probability of generating $a$ given $(p, q)$:

$$\mathcal{L}_{\text{SFT}} = -\mathbb{E}_{(p,q,a)\sim\mathcal{D}} \sum_{t=1}^{T} \log \pi_\theta(a_t \mid p, q, a_{<t}), \quad (1)$$

where $\mathcal{D}$ denotes the AesRecon dataset, $a = \{a_t\}_{t=1}^{T}$ is the token sequence of the editing actions, and $\pi_\theta$ is the token distribution of AesThinker.

## 4.2. Stage 1(b): GRPO-A

While SFT offers a strong initialization, relying on it alone often leads to overfitting to annotation patterns, limiting exploration and weakening generalization beyond the training distribution (Li et al., 2025d; Hao et al., 2025). This problem is even more pronounced in APR: unlike single-answer tasks (e.g., math problems or classification), photographic aesthetics is inherently multi-solution. For the same scene, different action plans (e.g., framing, viewpoint, pose, or depth-of-field) may all yield high-quality reconstructions, as shown in Figure 3. Therefore, supervision based on a single trajectory is intrinsically incomplete.

We adopt a fully on-policy variant of Group Relative Policy Optimization (GRPO) (Shao et al., 2024), named **GRPO-A**. Given a problem instance with a poor image $p$ and an instruction prompt $q$, we sample a group of $N$ action sequences $\{a_1, \ldots, a_N\}$ from the current policy $\pi_\theta(\cdot \mid p, q)$, and update the policy by maximizing:

$$\mathcal{J}(\theta) = \mathbb{E}\Big[\frac{1}{\sum_{i=1}^{N}|a_i|} \sum_{i=1}^{N} \sum_{j=1}^{|a_i|} \big(A_{i,j} - \beta D_{\text{KL}}\big[\pi_\theta \| \pi_{\text{ref}}\big]\big)\Big], \quad (2)$$

where $A_{i,j}$ is the advantage derived from the final rewards $\{r_1, \ldots, r_N\}$ of trajectories within the same group:

$$A_{i,j} = \frac{r_i - \text{mean}\big(\{r_k\}_{k=1}^{N}\big)}{\text{std}\big(\{r_k\}_{k=1}^{N}\big)}. \quad (3)$$

However, unlike many tasks where correctness can be verified by explicit rules (Guo et al., 2025), APR lacks an objective, computable signal to judge the quality of an action plan. Accordingly, we apply a pretrained MLLM (Bai et al., 2025) as a training-free reward model to evaluate sampled action sequences. Three complementary rewards are designed: a format reward $R_f$, an alignment reward $R_a$, and a creativity reward $R_c$.

**Format Reward ($R_f$).** Following prior work (Lin et al., 2025; Guo et al., 2025), we use a format reward $R_f \in \{0, 1\}$ to enforce structured outputs. We set $R_f = 1$ iff (i) all seven dimensions are present and each is enclosed by its corresponding special tokens, and (ii) the dimensions strictly follow the order $(1) \rightarrow (7)$; otherwise, $R_f = 0$.

**Alignment Reward ($R_a$).** Since the correctness of editing actions is difficult to measure with explicit, computable rules, we define an alignment reward $R_a \in [0, 5]$ based on semantic consistency. Specifically, we feed the reward model the sampled action sequence $a_i$ and the ground-truth action sequence $a$, and ask it to output a score $R_a(a_i, a) \in [0, 5]$ indicating whether they express the same editing intent

and key operations, regardless of surface wording. This reward encourages the policy to stay semantically aligned while allowing diverse yet equivalent wordings.

**Creativity Reward ($R_c$).** To encourage broader exploration of diverse action plans with higher potential for aesthetic improvement, we introduce a creativity reward $R_c$. Specifically, the reward model evaluates whether $a_i$ is (i) actionable and (ii) likely to produce a perceptible aesthetic gain for the poor image $p$, and outputs a score $R_c(p, a_i) \in [0, 5]$. To prevent unconstrained edits, we enforce a hard constraint: the action plan may only operate on subjects and scene elements present in the input image $p$, and must not introduce, assume, or describe any new objects, people, or background details. Any violation is assigned a low score.

Reading only the discrete score from the MLLM (Peng et al., 2025; Xiao et al., 2026) output often yields a coarse and sparse signal, and it ignores the model's uncertainty (Wu et al., 2023; Li et al., 2025c;d; Wang et al., 2025). Instead, we compute a logit-based score by taking the expected value over the score tokens:

$$R(\mathbf{X}) = \sum_{s \in \mathcal{O}} v(s) \cdot p(s \mid \mathbf{X}), \qquad (4)$$

where $\mathbf{X}$ denotes the reward-model input, $\mathcal{O}$ is the set of score tokens, $p(s \mid \mathbf{X})$ is the normalized probability of token $s$, and $v(s)$ maps token $s$ to its numerical value. We then normalize scores to the range $[0, 1]$:

$$\bar{R}(\mathbf{X}) = \frac{R(\mathbf{X}) - \min_{s \in \mathcal{O}} v(s)}{\max_{s \in \mathcal{O}} v(s) - \min_{s \in \mathcal{O}} v(s)}. \qquad (5)$$

We combine the three rewards as:

$$r_i = \lambda_f R_f(a_i) + \lambda_a R_a(a_i, a) + \lambda_c R_c(p, a_i), \qquad (6)$$

where $\lambda_f$, $\lambda_a$, and $\lambda_c$ control the contributions of formatting, semantic alignment, and creativity, respectively. Together, GRPO-A encourages exploration over diverse yet valid action plans, allowing AesThinker to surpass single-trajectory imitation learned from SFT.

### 4.3. Stage 2: Action-conditioned Editor Training

In Stage 2, we train **AesEditor**, an action-conditioned image editor, to execute structured editing actions and reconstruct a photo with improved aesthetics. However, existing instruction-following editors often produce edits that are semantically plausible yet aesthetically weak, and often struggle to ground photography-specific terminology into consistent pixel-level transformations. We therefore fine-tune an off-the-shelf image editing model (Wu et al., 2025a) on strictly aligned triplets $(p, g, a)$ from AesRecon, where $p$ is the poor image, $g$ is the aligned good reference, and $a$ is the ground-truth editing actions distilled in Stage 1.

AesEditor follows the flow-matching (rectified-flow) formulation of the base editor (Lipman et al., 2022; Liu et al., 2022), learning an action-conditioned velocity field that supports ODE-style sampling at inference. For each $(p, g, a)$, a frozen multimodal encoder produces conditioning states $h = \phi(p, a)$. The target image $g$ is encoded into a VAE latent $x_0$, and a noise latent $x_1 \sim \mathcal{N}(0, I)$ is sampled. With a timestep $t \in [0, 1]$, the interpolated latent is defined as

$$x_t = t x_0 + (1 - t) x_1, \qquad v_t = \frac{dx_t}{dt} = x_0 - x_1. \quad (7)$$

The MMDiT (Esser et al., 2024) backbone $v_\psi$ is trained to predict the target velocity from $(x_t, t, h)$ by minimizing

$$\mathcal{L}_{\text{edit}} = \mathbb{E}_{(p,g,a),\, x_1,\, t}\left[\left\|v_\psi(x_t, t, h) - v_t\right\|_2^2\right]. \qquad (8)$$

This yields an editor that faithfully executes high-level action plans as consistent pixel-level transformations, bridging aesthetic planning (Stage 1) and photo reconstruction.

**Inference.** At test time, we run a two-stage pipeline: (1) Aesthetic planning: AesThinker infers a structured action plan $\hat{a}$ from the input photo $p$ and the instruction prompt. (2) Aesthetic editing: conditioned on $(p, \hat{a})$, AesEditor generates the reconstructed output $\tilde{g} = \text{AesEditor}(p, \hat{a})$.

## 5. Experiments

### 5.1. Implementation Details

**Training.** We use Qwen3-VL-8B (Yang et al., 2025) and Qwen-Image-Edit-2511 (Wu et al., 2025a) as the base models for Stage 1 and Stage 2, respectively. In Stage 1(a), we use LoRA-based fine-tuning (Hu et al., 2022). In Stage 1(b), we perform GRPO-A with full-parameter fine-tuning, using Qwen2.5-VL-32B (Bai et al., 2025) as the reward model and setting $\lambda_f$, $\lambda_a$, and $\lambda_c$ to 0.1, 0.5, and 0.4, respectively. In Stage 2, we freeze the multimodal encoder and VAE, and apply LoRA only to the MMDiT. All experiments are conducted on 10 NVIDIA A40 GPUs (48GB each).

**Evaluation.** Recent editing benchmarks such as ImgEdit-Bench (Ye et al., 2025) and GEdit-Bench (Liu et al., 2025a) commonly use GPT-4o (Hurst et al., 2024) for discriminative evaluation. Following this setup, we conduct experiments on AesRecon and use GPT-4o as the evaluator. For each output, inspired by (You et al., 2025), we perform two pairwise comparisons against the original *poor* image and the corresponding *good* image, and report win rates, i.e., the fraction of cases where the output is judged more aesthetic than the compared reference. We further conduct human verification under the same protocol with 15 participants on a subset to validate the reliability of the GPT-4o judgments and better reflect human aesthetic preferences. In addition, we report aesthetic scores predicted by three widely used

*Table 1.* Quantitative results on AesRecon. We report win rates against the input *poor* image and the paired *good* image (GPT-4o and human) and three aesthetic scores. **Thinker** denotes an external planner (**None**: no planner). Nano Banana Pro is evaluated on a random 10% test subset due to API cost. Best and second-best results are in **bold** and underlined, respectively.

| Image Editing Model | Thinker | Win Rate vs. Poor ↑ | | Win Rate vs. Good ↑ | | Aesthetic Score ↑ | | |
|---|---|---|---|---|---|---|---|---|
| | | GPT-4o *(0-100%)* | Human *(0-100%)* | GPT-4o *(0-100%)* | Human *(0-100%)* | ArtiMuse *(0-100)* | LAION-V2 *(1-10)* | Q-ALIGN *(1-5)* |
| **Proprietary Models** | | | | | | | | |
| Nano Banana Pro *(Google)* | None | 54.44 | **72.55** | 16.67 | 21.95 | **50.90** | 5.59 | 3.24 |
| **Open-source Image Editing Models** | | | | | | | | |
| FLUX.1 Kontext [Dev] (Labs et al., 2025) | None | 12.96 | 5.88 | 2.66 | 3.66 | 38.34 | 5.07 | 2.83 |
| | Qwen3 | 11.96 | 7.84 | 2.88 | 4.88 | 38.60 | 5.08 | 2.80 |
| | GPT-4o | 12.51 | 13.72 | 3.88 | 6.10 | 38.10 | 5.05 | 2.76 |
| Bagel (Deng et al., 2025) | None | 12.40 | 17.65 | 7.75 | 12.20 | 37.69 | 4.94 | 2.58 |
| | Qwen3 | 3.43 | 19.61 | 2.32 | 6.10 | 35.18 | 4.74 | 2.39 |
| | GPT-4o | 7.31 | 21.57 | 2.99 | 7.32 | 33.99 | 4.65 | 2.33 |
| Step1X-Edit-v1.1 (Liu et al., 2025a) | None | 15.28 | 11.76 | 13.84 | 13.41 | 37.14 | 5.33 | 3.37 |
| | Qwen3 | 7.64 | 9.62 | 8.31 | 9.76 | 36.98 | 5.10 | 3.15 |
| | GPT-4o | 6.20 | 15.69 | 6.76 | 8.54 | 36.62 | 5.01 | 3.00 |
| Qwen-Image-Edit-2511 (Wu et al., 2025a) | None | 16.50 | 9.80 | 7.64 | 12.20 | 46.65 | 5.44 | 3.20 |
| | Qwen3 | 6.76 | 7.84 | 4.98 | 9.76 | 46.46 | 5.30 | 3.24 |
| | GPT-4o | 14.62 | 23.53 | 14.40 | 14.63 | 44.16 | 5.36 | 3.22 |
| **AesFormer (Ours)** | | **65.33** | 68.63 | **26.25** | **24.39** | 47.76 | **5.60** | **3.51** |

scorers, ArtiMuse (Cao et al., 2025), LAION-V2 (Schuhmann et al., 2022), and Q-ALIGN (Wu et al., 2023), to assess the aesthetic quality of the edited images.

### 5.2. Main Results

**Baselines.** To demonstrate the effectiveness of AesFormer, we compare against a set of competitive image editing models. Open-source baselines include FLUX.1 Kontext [Dev] (Labs et al., 2025), Bagel (Deng et al., 2025), Step1X-Edit-v1.1 (Liu et al., 2025a), and Qwen-Image-Edit-2511 (Wu et al., 2025a). We also evaluate Google's proprietary model, Nano Banana Pro, as a strong proprietary baseline; due to API cost, we evaluate it on a random 10% subset of the test set.

**Quantitative Results.** As shown in Table 1, consistent with prior findings (You et al., 2025), most existing open-source image editing models struggle in the APR setting: win rates are low against both the input *poor* image and the paired *good* image, indicating limited gains in structural aesthetics. In contrast, Nano Banana Pro performs substantially better, suggesting stronger out-of-the-box generalization. Our AesFormer consistently outperforms open-source baselines, achieving higher win rates and better scorers across the three aesthetic scorers. Notably, AesFormer matches and often outperforms Nano Banana Pro on most metrics, narrowing the gap to large-scale proprietary systems for APR.

This observation motivates a natural hypothesis: are open-source editors weak on APR primarily because they lack explicit editing instructions? Put differently, if we augment an Editor with a Thinker that generates edit actions, could APR be solved by simply composing an off-the-shelf Thinker with an off-the-shelf Editor? To test this, we construct two plug-and-play setups: we use Qwen3-VL-8B (Yang et al., 2025) and GPT-4o (Hurst et al., 2024) as the Thinker, generate actions with the same prompt used for AesThinker, and feed these actions as instruction inputs to different editors. As shown in Table 1, these off-the-shelf Thinker+Editor combinations do not produce consistent improvements and can even degrade performance for some editors.

Further analysis points to a two-sided mismatch. On the Thinker side, without APR-specific supervision, general-purpose planners often generate action plans that are vague, weakly goal-aligned, or underspecified for structural corrections, revealing limited understanding of photographic aesthetics. On the Editor side, most image editors offer limited support for aesthetic-driven structural edits and do not reliably ground photography terminology into faithful, executable transformations. Taken together, these results motivate AesFormer: an APR-specialized Thinker paired with a jointly aligned, action-conditioned Editor.

**Qualitative Results.** In Figure 4, we compare AesFormer with Qwen-Image-Edit-2511 (Wu et al., 2025a), Step1X-Edit-v1.1 (Liu et al., 2025a), and Nano Banana Pro. Most open-source models mainly focus on local appearance ad-

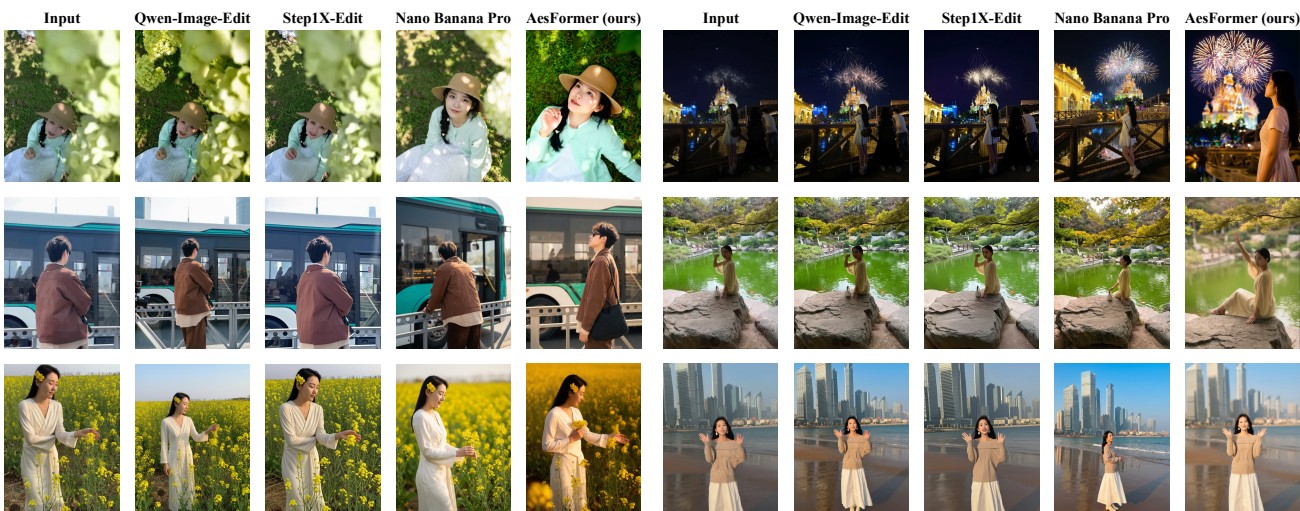

*Figure 4.* Qualitative comparison on AesRecon. Open-source editors often yield limited improvements in structural aesthetics, whereas AesFormer produces more consistent aesthetic enhancements and achieves results that are competitive with Nano Banana Pro.

*Table 2.* Ablation on AesRecon. We report performance after each training stage of AesFormer, where $S_{1a}$, $S_{1b}$, and $S_2$ denote Stage 1(a), Stage 1(b), and Stage 2, respectively. Best results are in **bold**.

| Settings | Win Rate vs. Poor ↑ | | Win Rate vs. Good ↑ | | Aesthetic Score ↑ | | |
|---|---|---|---|---|---|---|---|
| | GPT-4o *(0-100%)* | Human *(0-100%)* | GPT-4o *(0-100%)* | Human *(0-100%)* | ArtiMuse *(0-100)* | LAION-V2 *(1-10)* | Q-ALIGN *(1-5)* |
| Baseline (Edit-2511) | 16.50 ↑0.00 | 9.80 ↑0.00 | 7.64 ↑0.00 | 12.20 ↑0.00 | 46.65 ↑0.00 | 5.44 ↑0.00 | 3.20 ↑0.00 |
| $S_{1a}$ (shuffle) | 58.69 ↑42.19 | / | 18.60 ↑10.96 | / | 46.16 ↓0.49 | 5.49 ↑0.05 | 3.36 ↑0.16 |
| $S_{1a}$ | 61.04 ↑44.54 | / | 24.58 ↑16.94 | / | 47.70 ↑1.05 | 5.58 ↑0.14 | 3.48 ↑0.28 |
| $S_{1a} + S_2$ | 61.13 ↑44.63 | / | 24.14 ↑16.50 | / | 47.74 ↑1.09 | 5.58 ↑0.14 | 3.46 ↑0.26 |
| $S_{1a} + S_{1b} + S_2$ **(Ours)** | **65.33** ↑48.83 | **68.63** ↑58.83 | **26.25** ↑18.61 | **24.39** ↑12.19 | **47.76** ↑1.11 | **5.60** ↑0.16 | **3.51** ↑0.31 |

justments and struggle to execute the structural edits demanded by APR (e.g., resolving occlusions, reframing, and refining pose), often leaving capture-time structural issues unresolved. By contrast, AesFormer decouples aesthetic planning from editing: AesThinker diagnoses structural problems and produces actionable edit plans, which AesEditor then carries out as executable structural edits. This separation yields more reliable aesthetic improvements, producing results that are competitive with Nano Banana Pro. A qualitative comparison between AesThinker and GPT-4o (Hurst et al., 2024) is provided in Appendix.

### 5.3. Ablation Study

We conduct ablations of AesFormer, as shown in Table 2. The results show monotonic gains as stages are stacked, while stage removal causes a notable drop, indicating that the components are both necessary and complementary. We also test whether the ordered decision chain over seven photographic dimensions is essential. Specifically, we use Qwen2.5-VL-32B (Bai et al., 2025) to reorganize the action information in a shuffled order. This perturbation sub-

stantially degrades performance, suggesting that the model depends on both *what* to edit (the action set) and *when* to execute each decision (the order). The prescribed ordering thus acts as a key structural inductive bias, encouraging planning from global structure to local details.

## 6. Conclusion

This paper introduces **Aesthetic Photo Reconstruction (APR)**, which aims to improve a photo's aesthetic quality by correcting structural issues via reconstruction. We propose **AesFormer**, a two-stage framework that decouples aesthetic planning from image editing, with **GRPO-A** to encourage broader exploration over diverse action plans. To support APR research, we develop a video-based corpus-mining pipeline (**VCMP**) and build **AesRecon**, a new dataset and benchmark comprising 9,071 strictly aligned (poor, good) image pairs. Experiments show that AesFormer substantially improves APR performance, achieving results competitive with Google's Nano Banana Pro. Future work will extend APR to multi-image and video settings to further advance this line of research.

## Acknowledgements

This work was supported by the grants from the National Natural Science Foundation of China (62525201, 62132001, 62432001) and Beijing Natural Science Foundation (L247006).

## Impact Statement

This paper presents work whose goal is to advance the field of Machine Learning. There are many potential societal consequences of our work, none which we feel must be specifically highlighted here.

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
