# OpenReview forum: "AesFormer: Transform Everyday Photos into Beautiful Memories"
_ICML.cc/2026/Conference — ICML 2026 regular_

### Official Review · Reviewer_8jBy · 2026-02-23

**Soundness:** 2
**Presentation:** 2
**Significance:** 2
**Originality:** 3
**Overall Recommendation:** 3
**Confidence:** 4

**Summary:**

The paper introduces Aesthetic Photo Reconstruction (APR), a task aimed at correcting structural capture-time flaws in everyday photography (e.g., composition, pose, viewpoint) while preserving identity and scene semantics. The proposed system, AesFormer, is a two-stage framework: (1) AesThinker generates structured action plan. (2) AesEditor executes these actions through image editing models.
Planning is done by first training SFT, a model that outputs a sequence of edits to convert a bad picture to a good one, and then optimizing the output for diversity and better judgment using GRPO-A, a reinforcement learning model that can generate and ranks different edit sequences;
For supervision, the authors propose a video-based mining pipeline (VCMP), which segments tutorial videos temporally, samples frames at 2 FPS, and applies a ``good-frame detector'' to extract aligned poor/good pairs. The resulting dataset, AesRecon, contains ~9K image pairs.

**Compliance With Llm Reviewing Policy:**

Affirmed.

**Final Justification:**

Although the authors wrote a good rebuttal, I think that the paper still needs work and reassessment before publication so I will keep my score of weak reject.

**Key Questions For Authors:**

1. How sensitive is performance to 2FPS sampling and the good-frame detector?
2. How is the decision to treat the first frame of each segment as ``poor'' example correct?
3. How sensitive is the method to identity preservation?
2. How does the method handle personalization and aesthetic preferences?

**Limitations:**

See questions to authors

**Strengths And Weaknesses:**

Strengths:
APR moves beyond tone/style retouching toward structural correction, which is a meaningful direction.

Mining photography tutorials for weakly aligned ``poor→good'' examples is an elegant data collection strategy and addresses a real bottleneck.

The planner–editor split is conceptually reasonable and aligns with emerging trends in language-guided editing.

The two stage training of first using SFT and then optimizing using GRPO provides a good framework for solving such complex problem with no single solution.

Weaknesses:

1. Soundness
1.1. Arbitrary dataset construction choices: The 2FPS down-sampling and reliance on a good-frame detector permanently discard potentially informative frames. The paper does not provide sensitivity analysis or justification for these arbitrary decisions.

1.2. Heuristic poor-image selection: Selecting the first frame of each segment as the ``poor'' example appears arbitrary and potentially misleading. There is no empirical validation that this reliably captures flawed states.

1.3. Evaluation rigor is insufficient: The task is inherently subjective, yet there is no properly designed user study. Only a subset of the test set is manually evaluated, with limited details on annotator count, agreement, or statistical significance. For aesthetic evaluation, this is a major weakness.

1.4. Judge-model entanglement: The pipeline and evaluation rely heavily on proprietary LLMs (GPT-4o, GPT-5.2, Gemini 3, Qwen variants, InternVideo2.5, etc.). Some of these models are used both in data curation and evaluation. This raises concerns about reproducibility and possible bias.

1.5. Personalization: SOTA image editing methods are challenged to preserve person identity under image manipulations. This aspect is not explicitly discussed through the manuscript while it is a central trait for APR.

Overall, the empirical evidence does not fully support the strength of the claims.


2. Presentation. The narrative is readable at a high level, but several aspects reduce clarity. The Prior Work section does not meaningfully introduce or contextualize SFT and GRPO, though they are central to the method. The boundary between scientific contribution and system orchestration is unclear.
The pipeline integrates many large models, making it difficult to isolate what is genuinely novel versus what is prompt engineering or model stacking.
The high-level idea is clear, but methodological positioning and clarity around contributions need improvement.

3. Significance: Although APR is a potentially important direction, the current contribution appears largely system-level integration of many foundation models rather than a clearly defined methodological advance. The heavy dependence on proprietary components further limits impact, as the system may not be reproducible or extensible by the community.

Additionally, the personalization problem central to aesthetic judgment is not addressed. The method implicitly optimizes toward a generic aesthetic target, which weakens its practical significance.

---

> ### Author Rebuttal · Authors · 2026-03-31
>
> # Response to Reviewer 8jBy
> Thank you for your comments. We answer all concerns below:
> 1. **Sensitivity Analysis**
>
>     * **2 FPS Sampling.** In tutorial videos, the final polished result is typically displayed for about 2 seconds, so 2 FPS already captures **2–3 duplicate frames** of the same good result. Compared with 4 FPS, the overlap of unique good frames exceeds **97%**, indicating that denser sampling brings only marginal gain while doubling the cost. Since VCMP aims to efficiently mine APR data from highly redundant videos, rather than exhaustively preserve every frame, 2 FPS already provides near-saturated coverage at much lower cost.
>
>     * **Good-frame Detector.** As discussed in Line 193, the good-frame detector distinguishes clean, photo-like good frames from instructional frames with subtitles, annotations, and camera UI. On 600 manually annotated samples, it achieves over **93%** accuracy. Replacing it with Gemini 2.5 Pro or Qwen3-VL still yields over **95%** overlap in the selected good frames, showing that good-frame selection is robust to the detector choice.
>
> 2. **Poor-image Selection**
>
>     This choice is not arbitrary. Each segment represents a local improvement process in which the photo evolves from an improvable state to a better one. Therefore, the first frame is naturally the closest to the pre-improvement state (Line 202, left column). Moreover, the Alignment Check step(Fig. 2; Line 202, right column) further filters out mismatched pairs. Manual inspection of 1,200 poor-good pairs shows that **98%** of selected first frames correctly correspond to the flawed state improved by the good frame, validating this design.
>
> 3. **Identity Preservation**
>
>     We add quantitative experiments on **identity preservation** and **scene consistency** using 100 photos with clear human faces. Editing actions are generated by AesThinker and executed by different editors.
>
>     * **Identity Preservation:** measured by **ArcFace** and **AdaFace** as facial similarity between input and reconstructed images (-1~1; higher is better).
>
>     * **Scene Consistency:** evaluated by GPT-5.2 as the rate of not introducing new objects absent from the input (0%~100%; higher is better).
>
>     * **Aesthetics:** evaluated by ArtiMuse, LAION-V2, and Q-ALIGN.
>
>     |Model|Identity (ArcFace)|Identity (AdaFace)|Scene (GPT-5.2)|ArtiMuse|LAION-V2|Q-ALIGN|
>     |:-:|:-:|:-:|:-:|:-:|:-:|:-:|
>     Step1X-Edit-v1.1|0.44|0.43|79%|41.40|5.37|3.66|
>     Bagel|0.46|0.48|88%|41.84|5.48|3.42|
>     Nano Banana Pro|0.65|**0.66**|**94%**|43.36|5.75|3.62|
>     **AesFormer (ours)**|**0.72**|**0.66**|**94%**|**47.52**|**5.85**|**3.78**|
>
>     These results show that AesFormer preserves identity and scene better, and performs best on all three aesthetic metrics.
>
> 4. **Personalization and Aesthetic Preferences**
>
>     Although AesFormer mainly targets common aesthetic goals (Appendix C), it naturally supports personalization in two complementary ways.
>
>     **Way 1: Training-free Personalization**
>
>     Users can directly revise the editing actions generated by AesThinker in any dimension (e.g., composition, viewpoint, or pose) and feed the modified actions to AesEditor for preference-aware APR. To verify this, we revise the actions across all 7 dimensions on 100 photos to simulate different preferences, and evaluate whether the outputs follow the intended needs:
>
>     |Dimension|Rat.|Comp.|View.|Subj.|Pose|Foc.|C&L|Mean|
>     |-|:-:|:-:|:-:|:-:|:-:|:-:|:-:|:-:|
>     |Success Rate (%)|94|92|95|91|93|92|92|92.7|
>
>     The average success rate reaches **92.7%**, showing that AesFormer can reliably support personalized control across diverse dimensions even without additional training.
>
>     **Way 2: Lightweight Personalized Fine-tuning**
>
>     We study a niche moody cinematic preference using small personalized data of 157 poor-good image pairs (107 for training), and fine-tune AesThinker only:
>
>     |Setting|before|after|
>     |-|:-:|:-:|
>     |Success Rate (%)|4|92|
>
>     The success rate reaches **92%** after fine-tuning, showing that AesFormer can be effectively adapted to personalized preferences using only a small amount of data.
>
> 5. **User Study**
>
>     The human evaluation involves 15 participants under a unified protocol for fair comparison across all methods. The results are consistent with the main findings and further support the effectiveness of AesFormer. We will include more details on the protocol and analysis in the final version.
>
> 6. **Judge-model Entanglement & Reproducibility**
>
>     We clarify that evaluation uses only one proprietary model, GPT-4o, which is common in editing benchmarks (Line 311, right column). During data construction, GPT-4o is used only to filter failed overlay-removal cases (Line 197, right column), a step that is unrelated to the final preference evaluation and thus does not bias the evaluation. **We will release the dataset, benchmark, code, and checkpoints to support reproducibility and facilitate future research.**

---

> > ### Author Rebuttal · Reviewer_8jBy · 2026-04-03
> >
> > I appreciate the authors rebuttal. They did try to address many of my concerns.
> > Similar to other reviewers' comments I still have concerns regarding identity preservation and personalization.

---

> > > ### Author Response · Authors · 2026-04-04
> > >
> > > Thank you for the follow-up comment. We respectfully note that both concerns were addressed in **Points 3 and 4 of our previous rebuttal**, and we briefly restate the key evidence here for clarity.
> > >
> > > * **Identity Preservation (Point 3).** We added quantitative experiments on 100 face images using ArcFace, AdaFace, and GPT-5.2 scene consistency. AesFormer achieved strong identity and scene preservation (ArcFace **0.72**, AdaFace **0.66**, Scene **94%**), while also performing best on all three aesthetic metrics. These results show that AesFormer preserves identity and scene well while delivering stronger aesthetic reconstruction.
> > >
> > > * **Personalization (Point 4).** We clarified that AesFormer supports personalization in two complementary ways. (1) First, in a training-free setting, users can directly revise AesThinker’s editing actions, achieving a **92.7%** average success rate across seven dimensions. (2) Second, with lightweight personalized fine-tuning, the success rate improves from **4%** to **92%** using only 107 training pairs for a niche preference. These results demonstrate that AesFormer supports both controllable and data-efficient personalization.
> > >
> > > We hope this additional clarification helps make our previous rebuttal clearer.

---

### Official Review · Reviewer_zDKu · 2026-03-07

**Soundness:** 3
**Presentation:** 3
**Significance:** 2
**Originality:** 2
**Overall Recommendation:** 4
**Confidence:** 4

**Summary:**

This paper proposes a novel problem formulation called Aesthetic Photo Reconstruction (APR), which aims to improve aesthetic quality by enhancing portrait composition, camera viewpoint, or pose. To realize APR, the authors created a benchmark called AesRecon, constructed by automatically extracting poor and good scenes from video collected from online video platforms. The authors propose a method called AesFormer, which consists of AesThinker and AesEditor. AesThinker first plans how to edit the image. AesThinker is supervised fine-tuned using ground truth from AesRecon, then trained using GRPO-A to generate diverse plans. Based on AesThinker's plan, AesEditor edits the input image. Experiments compare the edited results with poor and good photos from AesRecon. Evaluations were conducted using GPT-4o and human assessment. Additionally, edited results were evaluated using three aesthetic scores. The evaluation results clearly show that AesFormer achieves higher performance than existing open-source models and can achieve performance comparable to Nano Banana Pro. An ablation study demonstrates the effectiveness of GRPO-A and the importance of fine-tuning AesEditor with AesRecon.

**Compliance With Llm Reviewing Policy:**

Affirmed.

**Final Justification:**

I would like to thank the authors for their responses. Since they provided thoughtful answers to my questions, I have upgraded my rating to "weak accept".

Regarding the authors' final response on content consistency, it is clear that AesFormer shows lower content consistency than real photographs. I strongly hope that the authors can provide further justification for why this weakness is not critical.

**Key Questions For Authors:**

Please see the Weaknesses.

**Limitations:**

yes

**Strengths And Weaknesses:**

**Strengths**
1. The authors have achieved an APR capable of improving portrait composition, camera viewpoint, or pose. Existing methods for enhancing portrait aesthetics primarily rely on retouching, which can only adjust overall image brightness and color tone. The proposed AesFormer enables more flexible editing, achieving further aesthetic improvements.

2. The authors created a new dataset, AesRecon. This plays a crucial role in enabling AesFormer to learn "how editing images improves aesthetic quality."

3. The authors propose GRPO-A to train AesThinker. Using GRPO-A has been shown to achieve higher performance than supervised fine-tuning alone.

4. The authors conducted experiments using multiple open-source models and Nano Banana Pro to evaluate AesFormer. They also performed ablation studies to demonstrate the effectiveness of GRPO-A and the importance of fine-tuning AesEditor with AesRecon. This evaluation is comprehensive, with compelling experiments conducted to demonstrate the effectiveness of the proposed method.


**Weaknesses**
1. The discussion lacks sufficient emphasis on the importance of the AesRecon dataset. Recent image editing datasets, such as Pico-Banana-400K [1] and HQ-Edit [2], create before-and-after pairs using image editing models. The authors should clarify how the method for creating such image editing datasets using image editing models differs from the method used to create AesRecon. For example, the PPR10K [3] dataset contains high-quality portraits. Could this be used as a dataset where these are considered good samples, and results from randomly editing composition or viewpoint using Qwen-Image-Edit2511 are considered poor samples?

[1] Qian, Yusu, et al. "Pico-banana-400k: A large-scale dataset for text-guided image editing." arXiv preprint arXiv:2510.19808 (2025).
[2] Hui, Mude, et al. "Hq-edit: A high-quality dataset for instruction-based image editing." arXiv preprint arXiv:2404.09990 (2024).
[3] Liang, Jie, et al. "Ppr10k: A large-scale portrait photo retouching dataset with human-region mask and group-level consistency." Proceedings of the IEEE/CVF Conference on Computer Vision and Pattern Recognition. 2021.

2. Discussion regarding research on improving aesthetics using aesthetic evaluation models is insufficient. For example, ImageReward [4] improves the quality of generated images using an aesthetic evaluation model, and this method does not require paired datasets. Similarly, it is conceivable that AesFormer could be trained using reinforcement learning with the scores from an aesthetic evaluation model as the reward. The authors should clarify the differences between such training methods and the training method using AesRecon.

[4] Xu, Jiazheng, et al. "Imagereward: Learning and evaluating human preferences for text-to-image generation." Advances in Neural Information Processing Systems 36 (2023): 15903-15935.

3. The authors adopt the flow: "aspect ratio $\rightarrow$ framing and composition $\rightarrow$ camera viewpoint $\rightarrow$ subject placement $\rightarrow$ subject pose and action details $\rightarrow$ focus and depth-of-field $\rightarrow$ color and light." However, the validation of this flow's effectiveness is insufficient. While Table 2 shows results for shuffled order, the authors should also demonstrate performance for other cases, such as free-form data or when the order is reversed.

4. The evaluation lacks the crucial criterion of "content consistency." For instance, if an object not present in the original photo appears after editing, it may negatively impact the user's impression. Similarly, loss of consistency in a person's appearance could adversely affect the user's perception. Evaluation should consider not only aesthetics but also content consistency.

5. According to Table 1, the proposed method's results significantly underperform compared to good photos. However, no analysis is provided explaining why it underperforms. The authors should present good photos in Figure 4 and discuss the differences between them and the proposed method's results.

---

> ### Author Rebuttal · Authors · 2026-03-31
>
> # Response to Reviewer zDKu
> Thank you for your comments. We answer all concerns below:
> 1. **Why AesRecon Matters**
>
>     * **Difference:** Existing datasets such as HQ-Edit mainly consist of **model-generated general editing pairs**. In contrast, AesRecon directly mines **real-world poor-good photo pairs** together with their improvement process from tutorial videos. It therefore offers authentic poor-to-good supervision for APR, rather than synthetic supervision from model-generated edits.
>
>     * **Importance:** This difference matters for three reasons.
>
>         * **Limited APR capability.** Model-generated before-after pairs are feasible for general edits (e.g., adding a hat), where existing models already perform well. In APR, however, current models still perform poorly (Table 1), making such a strategy unreliable for constructing poor-good photo pairs.
>
>         * **Synthetic defects do not reflect real photographic flaws and can hurt performance.** PPR10K focuses on retouching and contains strong tone and lighting, but its composition and viewpoint are often not optimal, making it a less suitable source of “good” images. At an early stage, we also explored constructing poor images by disrupting high-quality portraits with editing models. However, these defects were largely artificially introduced by the model rather than real photographic flaws. They failed to reflect real user mistakes or the diversity of real-world shooting problems, and incorporating such data noticeably hurt both the aesthetic scores and win rates of AesFormer.
>
>         * **Tutorial videos provide supervision unavailable in static image pairs.** Beyond before-after pairs, tutorial videos record the step-by-step improvement process from poor to good (Fig. 3). We find this process critical for producing reliable editing actions, whereas relying on image pairs alone often leads MLLMs to generate inaccurate actions.
>
> 2. **Why ImageReward-Style RL Is Not a Substitute**
>
>     * ImageReward targets **text-to-image generation**: a reward model scores a generated image $I_t$ conditioned on a text prompt $t$, i.e., $S = R(t, I_t)$, and this scalar reward is used to optimize the generator via RL.
>
>     * AesFormer instead addresses APR, essentially an **image editing** task. If RL were applied to optimize AesEditor, the editor would still require both a poor image $I_p$ and a text instruction (editing action) $t_e$ to generate the edited image $I_g$, with reward defined as $S = R(I_g, I_p, t_e)$. The key issue is that $t_e$ is not given, but must first be generated by AesThinker. Without poor-good pairs, AesThinker cannot be trained to produce appropriate $t_e$ for each poor image, so the RL pipeline for AesEditor cannot even be properly formed. Moreover, RL provides only outcome-level supervision, whereas AesRecon provides the paired, action-level supervision that is central to AesFormer.
>
> 3. **Effectiveness of the Proposed Flow**
>
>     We further evaluate a reversed-order setting, which degrades all three aesthetic metrics: ArtiMuse 47.49, LAION-V2 5.56, and Q-ALIGN 3.46. This further supports the effectiveness of the proposed flow for APR.
>
> 4. **Content Consistency**
>
>     We add quantitative experiments on **identity preservation** and **scene consistency**. Specifically, we randomly sample 100 photos with clear human faces, use AesThinker to generate editing actions, and reconstruct them with different editing models.
>
>     * **Identity Preservation:** measured by two representative face recognition models, **ArcFace** and **AdaFace**, which compute facial similarity between the input and reconstructed images (-1~1; higher is better).
>
>     * **Scene Consistency:** evaluated by GPT-5.2 as the rate of not introducing new objects absent from the input (0%~100%; higher is better).
>
>     * **Aesthetics:** evaluated by ArtiMuse, LAION-V2, and Q-ALIGN.
>
>     |Model|Identity (ArcFace)|Identity (AdaFace)|Scene (GPT-5.2)|ArtiMuse|LAION-V2|Q-ALIGN|
>     |:-:|:-:|:-:|:-:|:-:|:-:|:-:|
>     Step1X-Edit-v1.1|0.44|0.43|79%|41.40|5.37|3.66|
>     Bagel|0.46|0.48|88%|41.84|5.48|3.42|
>     Nano Banana Pro|0.65|**0.66**|**94%**|43.36|5.75|3.62|
>     **AesFormer (ours)**|**0.72**|**0.66**|**94%**|**47.52**|**5.85**|**3.78**|
>
>     These results show that AesFormer better preserves identity and scene, while outperforming all baselines on the three aesthetic metrics, yielding more reliable APR.
>
> 5. **Gap to Good Photos**
>
>     The good photos in AesRecon are high-quality targets that have been carefully refined, making them inherently difficult to surpass when reconstructing from a single poor photo. This also reflects the high quality of AesRecon. Nevertheless, AesFormer achieves higher Win Rate vs. Good than Nano Banana Pro under both GPT-4o and human evaluation, indicating that it is closer to these high-quality targets. We will add the paired good photos to Fig. 4 and further analyze their differences from AesFormer’s results in the final version.

---

> > ### Author Rebuttal · Reviewer_zDKu · 2026-04-02
> >
> > I would like to thank the authors for their response.
> > While the authors' response has addressed some of my concerns, I still have concerns regarding content consistency.
> >
> > I understand that AesFormer achieves higher content consistency than Nano Banana Pro, but it is unclear just how good these scores actually are.
> > For comparison, the authors should show the content consistency scores of the "good photos" included in AesRecon.
> > Since the "good photos" should depict exactly the same people and situations as the "input photos," content consistency between the two is considered an important reference.

---

> > > ### Author Response · Authors · 2026-04-04
> > >
> > > Thank you for reading our rebuttal. We agree that the consistency between the poor and good images in AesRecon can provide a useful reference for interpreting the reported preservation scores.
> > >
> > > However, we note that this reference is not suitable for direct computation over the full AesRecon dataset. APR does not require the subject appearance to remain strictly unchanged, and some poor–good pairs involve substantial yet photographically valid aesthetic edits. For example, some pairs involve large changes in viewpoint or pose (e.g., from a frontal face to a side or back view), changes in how accessories or props are presented (e.g., sunglasses or hats from handheld to worn), shallow depth of field that de-emphasizes the subject and highlights the scenery, or stronger rim lighting and different shadow patterns. These are common photographic adjustments and do not indicate a different identity or scene, but they can still reduce the measured consistency score.
> > >
> > > Therefore, to provide a fairer and more interpretable reference, we further evaluate identity and scene consistency on a curated subset of AesRecon that excludes such cases. The results are shown below:
> > >
> > > | Model | Identity (ArcFace) | Identity (AdaFace) | Scene (GPT-5.2) |
> > > |:-:|:-:|:-:|:-:|
> > > | Step1X-Edit-v1.1 | 0.44 | 0.43 | 79% |
> > > | Bagel | 0.46 | 0.48 | 88% |
> > > | Nano Banana Pro | 0.65 | 0.66 | 94% |
> > > | **AesFormer (ours)** | 0.72 | 0.66 | 94% |
> > > | **AesRecon** | **0.75** | **0.72** | **97%** |
> > >
> > > These results show that AesFormer substantially outperforms existing open-source baselines and achieves stronger or comparable consistency than Nano Banana Pro, while remaining closer to the AesRecon reference level across all metrics. We will include this analysis in the final version.

---

### Official Review · Reviewer_cZDU · 2026-03-11

**Soundness:** 2
**Presentation:** 3
**Significance:** 2
**Originality:** 3
**Overall Recommendation:** 4
**Confidence:** 4

**Summary:**

This paper proposes AesFormer, a framework to reconstruct photos to enhance their aesthetic quality. The framework consists of two stages: 1) A first stage that analyzes an input photo across seven progressive dimensions and plans the editing steps for the image, with an aesthetic action model AesThinker. Supervised fine-tuning (SFT) is applied to standardize the action format, and GRPO-A is adopted to explore more diverse and higher-quality action steps. 2) A second stage comprising an action-conditioned editor, AesEditor, which executes these actions to reconstruct the input image into an aesthetically pleasing photo. To support this research, the authors build a video-based corpus-mining pipeline (VCMP) that extracts poor and good image pairs from online photography tutorial videos, resulting in the dataset AesRecon. Experiments show that AesFormer improves in the aesthetic photo reconstruction task and is competitive with large-scale proprietary systems.

**Compliance With Llm Reviewing Policy:**

Affirmed.

**Final Justification:**

The rebuttal addressed my major concerns of content preservation and ablation study, providing a more comprehensive evaluation of the proposed framework. Therefore, I upgraded my score to weak accept.

**Key Questions For Authors:**

1. What was the specific reason for choosing GPT as the action generator and Gemini for action validation?
2. The quantitative results indicate that AesFormer is competitive with Nano Banana Pro. Could the authors further discuss the specific cases or technical aspects where AesFormer has the advantages compared to large-scale propriety systems?
3. Why were the human win rates omitted from Table 2?

**Limitations:**

Yes.

**Strengths And Weaknesses:**

Strengths:
1) This paper address the significant challenge of reconstructing original photo for better aesthetics through global adjustments (e.g., pose, framing, and lighting) rather than just local appearance edits. This setting is both more challenging and more applicable to real-world scenarios.
2) The framework proposed is technically sound and aligned with human photographic priors, from planning across multiple dimensions to the execution of those planned actions.
3) The new dataset curated from photographic tutorial videos addresses the scarcity of high-quality paired examples that include corresponding instructional text.
4) The paper is well-structured and clearly written.

Weakness:
1) Insufficient Evaluation. There is a lack of quantitative analysis of the identity preservation of the reconstructed photos. Providing visual examples within the ablation studies would better demonstrate the model's performance.
2) Dataset Bias. The results of the proposed AesRecon may be highly influenced by the dataset's specific style, as seen in Figure 4, exhibiting high saturation and artificial background blur, which is not inconsistent with the original input image.

---

> ### Author Rebuttal · Authors · 2026-03-31
>
> # Response to Reviewer cZDU
> Thank you for your comments. We answer all concerns below:
> 1. **Identity Preservation**
>
>     Thank you for your suggestion. We add quantitative experiments on **identity preservation** and **scene consistency**. Specifically, we randomly sample 100 photos with clear human faces, use AesThinker to generate editing actions, and reconstruct them with different editing models.
>
>     * **Identity Preservation:** measured by two representative face recognition models, **ArcFace** and **AdaFace**, which compute facial similarity between the input and reconstructed images (-1~1; higher is better).
>
>     * **Scene Consistency:** evaluated by GPT-5.2 as the rate of not introducing new objects absent from the input (0%~100%; higher is better).
>
>     * **Aesthetics:** evaluated by ArtiMuse, LAION-V2, and Q-ALIGN.
>
>     |Model|Identity (ArcFace)|Identity (AdaFace)|Scene (GPT-5.2)|ArtiMuse|LAION-V2|Q-ALIGN|
>     |:-:|:-:|:-:|:-:|:-:|:-:|:-:|
>     Step1X-Edit-v1.1|0.44|0.43|79%|41.40|5.37|3.66|
>     Bagel|0.46|0.48|88%|41.84|5.48|3.42|
>     Nano Banana Pro|0.65|**0.66**|**94%**|43.36|5.75|3.62|
>     **AesFormer (ours)**|**0.72**|**0.66**|**94%**|**47.52**|**5.85**|**3.78**|
>
>     These results show that AesFormer preserves identity and scene better, and performs best on all three aesthetic metrics, yielding more reliable APR.
>
> 2. **Dataset Bias and Visual Examples**
>
>     We respectfully clarify that the examples in Fig. 4 are for qualitative illustration and do not reflect the overall distribution of AesRecon. Our analysis shows that only **26%** of the photos exhibit high saturation and only **8%** show noticeable background blur, indicating that such effects are not common in the dataset. Moreover, background blur is often a natural photographic choice to emphasize the subject, rather than an unnatural artifact inconsistent with the original scene. In the final version, we will include more diverse qualitative examples, including ablation visualizations, and provide a more detailed distribution analysis over scene, subject, and style dimensions.
>
> 3. **Generator–Validator Choice**
>
>     This choice is based on empirical performance. We evaluate multiple generator–validator combinations and find that GPT-5.2 as the generator and Gemini 3 as the validator yields the most reliable results in terms of action quality and image–action alignment. Using a different model for validation also helps reduce self-confirmation bias.
>
> 4. **AesFormer vs. Proprietary Systems**
>
>     * **Specific Cases:** Compared with Nano Banana Pro, AesFormer tends to make larger yet still plausible adjustments to composition and human pose, likely because GRPO-A encourages broader exploration, as shown in Fig. 4 of the main paper. By contrast, Nano Banana Pro is generally more conservative and often prefers smaller, safer edits. Despite these larger adjustments, AesFormer still achieves comparable or better identity and scene preservation, while producing higher aesthetic scores.
>
>     * **Technical Advantages:** AesFormer offers three main advantages:
>         * **Modularity and Extensibility.** By explicitly decoupling aesthetic planning from image editing, AesFormer allows the two modules to be improved independently. For example, AesThinker can be upgraded separately, and AesEditor can be replaced as stronger editing models emerge, without redesigning the whole framework.
>
>         * **Interpretability and Controllability.** Unlike proprietary black-box systems, AesFormer exposes an intermediate aesthetic plan that users can inspect, edit, and customize before reconstruction. This also enables personalization, as discussed in our response to Reviewer x5tS (“3. Personalization and User Preferences,” Way 1).
>
>         * **Efficiency and Personalization.** Compared with large proprietary systems, AesFormer is smaller and more computationally efficient. It can also be adapted to private preferences with a small amount of data through lightweight fine-tuning, as discussed in our response to Reviewer x5tS (“3. Personalization and User Preferences,” Way 2).
>
> 5. **Human Win Rates in Table 2**
>
>     Human evaluation is expensive, so in the original submission we used it only in Table 1 for comparisons with other methods. Table 2 mainly aims to verify whether each training stage brings consistent gains, which is already supported by the GPT-4o win rates and the three aesthetic metrics. To make the ablation more complete, we now additionally add the human win rates below.
>
>     Settings|Human vs. Poor|Human vs. Good|
>     |:-:|:-:|:-:|
>     Baseline (Edit-2511)|9.80|12.20|
>     $S_{1a}$​ (shuffle)|54.90|17.07
>     $S_{1a}$|60.78|19.51
>     $S_{1a}$+$S_{2}$|58.82|18.29
>     $S_{1a}$​+$S_{1b}$​|64.71|21.95
>     **$S_{1a}$​+$S_{1b}$​+$S_{2}$​ (ours)**|**68.63**|**24.39**
>
>     The added human results are consistent with the original results and further confirm that each training stage contributes positively to the final performance.

---

> > ### Author Rebuttal · Reviewer_cZDU · 2026-04-02
> >
> > I would like to thank the authors for addressing my concerns regarding the evaluation. However, I still have the following concerns:
> > 1) **Content/Identity Preservation**: The scores for content and identity preservation are competitive with large-scale proprietary systems, but the improvement remains marginal. As it is believed that content preservation is critical in the APR task, I am interested in understanding which specific part of the proposed pipeline contributes to thiis performance. Or, is this primarily an inheritance of the inherent capabilities of the base Qwen model?
> > 2) **Human Evaluation**: I appreciate the authors’ further explanation of the model's personalization potential, as well as the inclusion of human judgment results in the responses to other reviewers, and the complete Table 2. However, since there are no visualization results for the ablation studies, the qualitative assessment remains limited. Furthermore, the current human evaluation appears quite subjective and lasks transparency. The lack of specific details regarding the number of participants and their expertise (laypeople vs. professionals), making the evaluation less convincing.

---

> > > ### Author Response · Authors · 2026-04-04
> > >
> > > 1. **Content/Identity Preservation**
> > >
> > >     Thank you for your comment. The strong preservation performance largely benefits from the pretrained Qwen-Image-Edit backbone. Within our pipeline, the component most directly related to preservation is **Stage 2 (AesEditor)** rather than Stage 1. Stage 1 (AesThinker) focuses on aesthetic understanding and editing-action planning, and thus does not directly affect preservation. In contrast, Stage 2 further trains Qwen-Image-Edit on AesRecon triplets of (poor image, editing actions, good image).
> > >
> > >     Importantly, we would like to clarify that the goal of Stage 2, and of this paper, is **not to improve preservation metrics in isolation**. Instead, we observe that most image editing models cannot reliably execute the professional editing actions generated by AesThinker—particularly those involving substantial changes in composition, viewpoint, camera angle, pose, and lighting—and therefore often produce only minor or ineffective edits (Line 81 of the main paper). In other words, **strong preservation alone is insufficient for APR if the model cannot perform the substantial edits required for aesthetic reconstruction**. To address this, we introduce Stage 2 to improve the model’s ability to faithfully execute such challenging edits, while preserving the backbone’s original identity and scene consistency as much as possible.
> > >
> > >     To verify this, we compare the original Qwen-Image-Edit and AesEditor under the same editing actions generated by AesThinker. We report two win rates, three aesthetic metrics, and identity/scene preservation results below:
> > >
> > >     |Model|Win Rate vs. Poor|Win Rate vs. Good|ArtiMuse|LAION-V2|Q-ALIGN|Identity (ArcFace)|Identity (AdaFace)|Scene (GPT-5.2)|
> > >     |:-:|:-:|:-:|:-:|:-:|:-:|:-:|:-:|:-:|
> > >     +Qwen-Image-Edit|62.79|24.70|47.67|5.59|3.49|0.71|**0.67**|92%|
> > >     +AesEditor **(ours)**|**65.33**|**26.25**|**47.76**|**5.60**|**3.51**|**0.72**|0.66|**94%**|
> > >
> > >     The results show that, compared with the original Qwen-Image-Edit, **AesEditor achieves better aesthetic reconstruction**, as reflected by improvements in both win rates and all three aesthetic metrics, while **maintaining identity and scene preservation**. These results show that Stage 2 enables more effective APR editing without compromising the backbone’s preservation ability, which is exactly the goal of our Stage 2 design.
> > >
> > > 2. **Ablation Visualization & Human Evaluation**
> > >
> > >     * **Ablation Visualization:** Thank you for your suggestion. We agree that visualizing the ablation results can provide a more direct understanding of how each component affects reconstruction quality. To address this, we have added qualitative visualizations of the ablation studies in an anonymous repository: https://anonymous.4open.science/r/visualization-results-for-the-ablation-studies/visualization-results-for-the-ablation-studies.png. These visual results are consistent with the quantitative findings in the paper. AesFormer achieves the best reconstruction quality among the compared settings, while removing any component leads to clear degradation, further confirming the effectiveness of each component qualitatively.
> > >
> > >     * **Human Evaluation:** We clarify the human evaluation protocol. Specifically, 15 evaluators participated in the study, all with at least an undergraduate-level educational background. Among them, 12 were general users and 3 were photography enthusiasts with relevant experience. Before the formal evaluation, we conducted a standardized instruction and calibration process using examples outside the test set to establish a shared understanding of the assessment criteria and improve rating consistency. In addition, the evaluation was conducted in a blind setting, where evaluators did not know which model produced each result, to reduce potential subjective bias.
> > >
> > >     We will include both the ablation visualizations and a more detailed description of the human evaluation protocol in the final version.

---

### Official Review · Reviewer_x5tS · 2026-03-12

**Soundness:** 3
**Presentation:** 3
**Significance:** 3
**Originality:** 3
**Overall Recommendation:** 5
**Confidence:** 3

**Summary:**

The paper presents AesFormer, a two-stage framework for enhancing the aesthetics of non-professional photos.
The idea behind this framework is to address the problem in two separate steps: in the first step, a structured plan of actions to enhance the aesthetic of the photo is defined, while in the second step, the editing is performed. To train this framework, a large dataset of poor and good couples of images, extracted from video tutorials and other online sources, has been created.
An extensive analysis of the performance of the proposed framework has been performed using aestetic metrics, human assessment and  using GPT-4o as the evaluator. Finally ana blation study is also performed revealing the importance of all the module end in particular of the order of the editing actions.

**Compliance With Llm Reviewing Policy:**

Affirmed.

**Final Justification:**

The authors properly addressed my concerns in the rebuttal phase. In particular thay added a section related to the computational costs, and discussions on user-personalization.
I confirm my previous reccomendation.

**Key Questions For Authors:**

1) The framework relies on several computationally intensive models. The resources required to train and apply the framework are not discussed, but it could be a limit in its adoption.
2) It is unclear whether the framework and dataset will be made available to the research community.
3) Could the framework be fine-tuned to follow user preferences instead of relying solely on common rules collected from the web?

**Limitations:**

The limitations are discussed in the appendix, but this information is important enough to be included in the main paper.
Another important issue that should be discussed is the computational cost of training the model and processing the photo, as this could limit the adoption of the model for instance in mobile applications.

**Strengths And Weaknesses:**

The paper has several strengths. First, it offers a creative yet well-structured and theoretically grounded framework for enhancing the aesthetic appeal of non-professional photos.
Secondly, it provides a substantial dataset comprising pairs of images, one representing a poor image and the other the aesthetically enhanced version. Where available, text describing the steps taken to achieve the result is also included. Another important contribution of this work is the focus on increasing the creativity of the framework, including rewards to allow for a greater variety of outcomes.
Another strength of the paper is the structured and comprehensive analysis of the results from quantitative and qualitative perspectives. The paper is well-written and clear. However, the use of several acronyms in the same sentence sometimes makes it slightly difficult to understand.
A potential weakness of the proposed framework is that it may be extremely challenging to reproduce due to its complexity if the code and dataset are not made available.

---

> ### Author Rebuttal · Authors · 2026-03-31
>
> # Response to Reviewer x5tS
> Thank you for your comments. We answer all concerns below:
> 1. **Computational Cost**
>
>     All experiments are conducted on **10 NVIDIA A40 GPUs (48GB each)**. To further clarify the resource requirements of AesFormer, we summarize the main training and inference costs below.
>
>     |Phase|Component|Resources|Time|
>     |-|-|-|-|
>     |Train|AesFormer Stage 1(a)|8 GPUs|3 h|
>     |Train|AesFormer Stage 1(b)|8 GPUs for GRPO-A + 2 GPUs for the reward model|95 h|
>     |Train|AesFormer Stage 2|8 GPUs|24 h|
>     |Test (single-sample inference)|AesThinker|1 GPU|~16 s|
>     |Test (single-sample inference)|AesEditor|1 GPU|~2 min|
>
>     Overall, the main cost of AesFormer lies in training, while test-time inference is feasible on a single GPU. We will include these details in the final version for a clearer presentation of its computational requirements.
>
> 2. **Public Release**
>
>     We commit to publicly releasing the AesRecon dataset, benchmark, code, and checkpoints to facilitate future research.
>
> 3. **Personalization and User Preferences**
>
>     Thank you for your valuable suggestion. We agree that personalization is an important extension of APR. Although AesFormer mainly targets common aesthetic goals (Appendix C), its decoupled design naturally supports personalization in two complementary ways.
>
>     **Way 1: Training-free Personalization**
>
>     * **Method:** Users can directly revise the editing actions generated by AesThinker in different dimensions (e.g., composition, viewpoint, and pose) and feed the modified actions to AesEditor for preference-aware reconstruction. Because these actions are transparently exposed to users and explicitly organized by progressive photographic dimensions, users can easily identify and adjust the aspects they care about.
>     * **Experiments:** To evaluate this setting, we conduct experiments on 100 photos. We first generate default reconstruction results with AesFormer, then manually revise the editing actions in each of the seven dimensions to simulate different personalized preferences, and finally use AesEditor to reconstruct the photos. Human evaluators assess whether the outputs satisfy the specified preferences. The results are shown below.
>
>         |Dimension|Rat.|Comp.|View.|Subj.|Pose|Foc.|C&L|Mean|
>         |-|:-:|:-:|:-:|:-:|:-:|:-:|:-:|:-:|
>         |Success Rate (%)|94|92|95|91|93|92|92|92.7|
>
>     * **Conclusion:** The average success rate reaches **92.7%**, showing that AesFormer can reliably support personalized control across diverse photographic dimensions even without additional training.
>
>     **Way 2: Lightweight Personalized Fine-tuning**
>
>     * **Method:** AesFormer can also be adapted to user-specific preferences by fine-tuning on a small amount of private data.
>     * **Experiments:** We study a niche preference, namely a moody cinematic aesthetic. Specifically, we construct a small personalized dataset with **157 poor-good image pairs** (107 for training and 50 for testing) that reflects this preference, and perform standard SFT on **AesThinker only**. The success rates before and after fine-tuning are reported below. The “before fine-tuning” result is obtained under AesFormer’s default setting.
>
>         |Setting|before fine-tuning|after fine-tuning|
>         |-|:-:|:-:|
>         |Success Rate (%)|4|92|
>
>     * **Conclusion:** After fine-tuning, the success rate reaches **92%**, showing that AesFormer can be effectively adapted to personalized aesthetic preferences with only a small amount of private data.
>
>     Overall, AesFormer supports personalization well under both training-free personalization and lightweight fine-tuning. We will include more personalized visual examples in the final version to further demonstrate this capability.
>
> 4. **Limitations**
>
>     Thank you for the suggestion. We agree that the limitations are important and will move them from the appendix to the main paper in the final version.

---

> > ### Author Rebuttal · Reviewer_x5tS · 2026-04-04
> >
> > Thank you for your response and for addressing my concerns.

---

> > > ### Author Response · Authors · 2026-04-04
> > >
> > > Thank you for reading our rebuttal. We will carefully incorporate all the promised revisions into the final version to further improve the quality of the paper.

---

### Decision · Program_Chairs · 2026-04-30

**Decision:**

Accept (regular)

**Comment:**

The paper still has  limitations in evaluation rigor and dependence on external model-based supervision, but the rebuttal successfully addressed several major concerns, two reviewers increased their scores, and the paper offers a meaningful and novel task setting plus dataset and a reasonably convincing system baseline for that setting. Considering the pros and cons of the paper, I would lean toward acceptance even though one reviewer is still negative.